# Erk1/2-Dependent HNSCC Cell Susceptibility to Erastin-Induced Ferroptosis

**DOI:** 10.3390/cells12020336

**Published:** 2023-01-16

**Authors:** Dragana Savic, Teresa Bernadette Steinbichler, Julia Ingruber, Giulia Negro, Bertram Aschenbrenner, Herbert Riechelmann, Ute Ganswindt, Sergej Skvortsov, József Dudás, Ira-Ida Skvortsova

**Affiliations:** 1Laboratory for Experimental and Translational Research on Radiation Oncology (EXTRO-Lab), Department of Therapeutic Radiology and Oncology, Medical University of Innsbruck, A-6020 Innsbruck, Austria; 2Tyrolean Cancer Research Institute (TKFI), A-6020 Innsbruck, Austria; 3Department of Otorhinolaryngology and Head and Neck Surgery, Medical University of Innsbruck, A-6020 Innsbruck, Austria; 4University Hospital of Tyrol, A-6020 Innsbruck, Austria; 5Department of Dermatology, Medical University of Vienna, A-1090 Vienna, Austria; 6Department of Therapeutic Radiology and Oncology, A-6020 Innsbruck, Austria

**Keywords:** HNSCC, erastin, xCT, ERK signaling

## Abstract

Unfavorable clinical outcomes mean that cancer researchers must attempt to develop novel therapeutic strategies to overcome therapeutic resistance in patients with HNSCC. Recently, ferroptosis was shown to be a promising pathway possessing druggable targets, such as xCT (SLC7A11). Unfortunately, little is known about the molecular mechanisms underlying the susceptibility of HNSCC cells to ferroptosis. The goal of this study was to determine whether HNSCC cells with activated Erk1/2 are vulnerable to ferroptosis induction. Our results have shown that xCT (SLC7A11) was overexpressed in malignant tissues obtained from the patients with HNSCC, whereas normal mucosa demonstrated weak expression of the protein. In order to investigate the role of Erk1/2 in the decrease in cell viability caused by erastin, xCT-overexpressing FaDu and SCC25 HNSCC cells were used. The ravoxertinib-dependent inhibition of Erk1/2 signaling led to the decrease in erastin efficacy due to the effect on ROS production and the upregulation of ROS scavengers SOD1 and SOD2, resulting in repressed lipid peroxidation. Therefore, it was concluded that the erastin-dependent activation of ferroptosis seems to be a promising approach which can be further developed as an additional strategy for the treatment of HNSCC. As ferroptosis induction via erastin is strongly dependent on the expression of Erk1/2, this MAP kinase can be considered as a predictor for cancer cells’ response to erastin.

## 1. Introduction

Head and neck squamous cell carcinoma (HNSCC) is the sixth most common malignant tumor worldwide, with approximately 377,700 new cases and causing 177,800 deaths in 2020 [1]. As HNSCCs are highly heterogeneous tumors, they demonstrate different therapy responses and survival rates depending on the anatomical site, stage, and human papilloma virus (HPV) status. Thus, in the most recent EUROCARE study, an approximately 40% age-standardized 5-year relative survival rate was reported for HNSCC patients, with the poorest survival rate being associated with hypopharyngeal cancer (~25%) and the highest being associated with laryngeal cancer (~59%) [2]. Differences in survival rates are also dependent on the therapy responses of HNSCCs. It was observed that the best therapeutic outcomes were registered in patients with HNSCC with localized and loco-regional diseases compared to the patients with distant metastatic tumors. However, HPV-negative loco-regional tumors show a three-year survival rate of ~37% [3]. Despite the opportunities to use a variety of therapeutic innovations during the last few decades, treatment outcomes in patients with HNSCC remain mostly unchanged [3]. Hence, novel therapeutic strategies and compounds are needed to improve the therapy results and survival rates in patients with HNSCC.

Radiation therapy and the majority of chemotherapeutics are used in HNSCC management to induce carcinoma cell death. Furthermore, the ability of cancer cells to demonstrate treatment-caused cell death can be closely related to the levels of therapy responses and treatment outcomes in patients. Unfortunately, the mechanisms underlying the therapy resistance of HNSCC cells show a crosstalk with the molecular mechanisms inhibiting cell death. Luckily, carcinoma cells utilize different mechanisms regulating cell death, and if the most frequently used apoptosis or necroptosis are not effective enough in response to anti-cancer treatment, other pathways can be successfully activated. It is interesting to note that malignant cells can be susceptible to the recently discovered ferroptosis even upon evading other types of cell death [4].

Ferroptosis, as a distinct and special type of cell death, was first described in 2012 [5,6,7]. As ferroptosis is highly dependent on the intracellular iron concentration, and HNSCC cells very often demonstrate increased iron uptake, resulting in intracellular augmentation [8], it is possible to assume that ferroptosis induction in HNSCC cells is a prospective and promising therapeutic approach. Indeed, a number of scientific reports regarding the role and molecular mechanisms of ferroptosis in HNSCC cell death have been published [9,10,11]. Although the ferroptotic pathway is not fully understood, the inhibition of the key ferroptosis-related molecule cystine/glutamate antiporter (xCT) can lead to the abrogation of cancer cell resistance to chemo-radiotherapy [10,12,13,14,15]. Erastin is a potent xCT inhibitor, and its efficacy to induce ferroptosis and kill HNSCC cells was shown in preclinical studies [16,17,18].

It is currently known that ferroptosis induction is accompanied by intracellular ROS formation, and the level of ferroptosis can be suboptimal if the antioxidant system is upregulated [10]. One of the main regulators of intracellular ROS is the Nrf2 molecule [19,20,21,22,23]. Therefore, the inhibition of Nrf2 is associated with an improvement in the cellular response of squamous cell carcinoma cells to cytotoxic agents [10,11,19]. Furthermore, Nrf2 has been described to regulate ferroptosis in cooperation with mitogen-activating protein (MAP) kinases [24]. This observation allows one to assume that MAP kinases can be implicated in ferroptosis development via the regulation of intracellular ROS production. The overexpression of the epidermal growth factor receptor (EGFR) accompanied by the downstream activation of MAP kinases is observed in 90% of HNSCCs [25,26,27]. Indeed, Poursaitidis et al. (2017) demonstrated that cells can be protected from ferroptosis upon the inhibition of MAP kinases [28]. As a number of MAP kinases (Erk1/2, p38MAPK, and JNK1/2) are involved in the regulation of oxidative stress, only Erk1/2 is phosphorylated in an EGFR-dependent manner, whereas p38MAPK and JNK1/2 activation is executed as an EGFR-independent mode [29]. Phosphorylated Erk1/2 can be found in more than 90% of patients with HNSCC [27,30], and the activation of Erk1/2 is associated with the more aggressive behavior of the tumors, the increased development of recurrences, and advanced nodal stages [27,30,31]. Furthermore, activated Erk1/2 regulate and control a number of processes, such as cancer cell proliferation, cell death and survival, differentiation, and therapy resistance [32]. Taking into account the fact that MAP kinases, including Erk1/2, can be implicated in HNSCC cell insensitivity to anti-tumor therapeutic approaches, we aimed to determine whether HNSCC cells with activated Erk1/2 are vulnerable to ferroptosis induction and whether ferroptosis development can be modulated by the inhibition of Erk1/2 expression and phosphorylation.

## 2. Materials and Methods

### 2.1. Patients

In this study, we enrolled 37 patients diagnosed with HNSCC and treated at Innsbruck Country Hospital (Tirol Kliniken) in the period between February 2013 and February 2018 (Appendix A). The primary tumor biopsies were performed before therapy, and the evaluation of gene expression and immunohistopathological analysis were performed from these samples. Ethical permission to collect and analyze the tumor specimens was obtained from the Ethics Committee at Innsbruck Medical University (UN4428, date: 26 July 2011). Details of the patients’ characteristics are summarized in Appendix A. The study was extended by 10 normal mucosa samples from uvulopalatopharyngoplasty.

### 2.2. RNA Isolation and Relative Quantification via RT-PCR

For RNA isolation, 2–3 mm tissue slices were collected and lysed in 1 mL of TRIzol^®^ Reagent (Ambion^®^, Life technologies™, Carlsbad, CA, USA); RNA was isolated as instructed by the manufacturer. RNA concentrations were determined via fluorometric measurements (Qubit, Invitrogen, Darmstadt, Germany), and RNA quality and integrity were identified using the Qubit RNA IQ kit (Invitrogen). The proportion of intact RNA from the total RNA isolates was at least 75%. Two micrograms of total RNA were reverse transcribed with M-MuLV Reverse Transcriptase and two micrograms of oligo dT15 (GeneON, Ludwigshafen am Rhein, Germany) in a ThermoQ heating and cooling block (Biozym, Hessisch Oldendorf, Germany). Oligonucleotide primer sequences for xCT were SLC7A11 forward: 5′-CAAATGCAGTGGCAGTGACCTT-3′ and SLC7A11 reverse: 5′-ACCGTTCATGGAGCCAAAGC-3′ [33], and the loading control housekeeping gene glyceraldehyde-3-phosphate dehydrogenase (GAPDH) (forward: 5′-TGCACCACCAACTGCTTAGC-3′and reverse: 5′-GGCATGGACTGTGGTCATGAG-3′) [34]. The primers were synthesized by Invitrogen, Darmstadt, Germany. They were used for real-time PCR by utilizing the Sensifast Sybr Fluorescein Kit of Bioline (Labconsulting, Vienna, Austria) and the Bio-Rad MyiQ™ (Bio-Rad, Laboratories, Inc., Hercules, CA, USA) cycler according to the manufacturer’s protocol.

Relative quantities of SLC7A11 transcripts were calculated using pair-wise differences of threshold cycles (∆CT) of the gene of interest and the loading control housekeeping gene [35]. The CT was defined as the number of PCR cycles required for a sample to cross a threshold line as determined by previous negative controls. The result represented the ratio of the target gene expression to the stable internal loading control [36]. Ten control normal mucosa and thirty-seven HNSCC tumor tissue samples were included in the real-time PCR analysis by double pipetting each sample. The CT values of GAPDH in all the samples were from 20 to 22 and did not show sample-specific regulation. The CT values of SLC7A11 ranged from 26 to 33 and showed significant differences between the control and HNSCC samples.

### 2.3. Immunohistochemistry and Immunocytochemistry Staining and Image Acquisition

Enzyme immunohistochemistry (IHC) for xCT was carried out using the mouse monoclonal anti-xCT antibody (ab37185; Abcam, Cambridge, UK), and the reaction was developed using a universal secondary antibody (Roche Ventana, Mannheim, Germany) and the DAB Map kit of Ventana.

The immunohistochemistry labeling levels were acquired in the TissueFaxs system in brightfield using a firewire-connected Pixelink camera (Pixelink, Rochester, NY, USA). The acquired images were imported into the HistoQuest program, and the staining signal was quantified using a single-reference-shade color deconvolution algorithm [16]. The staining signal was a mixture of a brown reaction and a blue counterstain of hematoxylin, which could also present itself as a clear blue color or clear brown color as completely negative or fully positive reactions. The analysis was performed using the HistoQuest software by counting positive cells (brown) in relation to all cells recognized based on “blue cell nuclei.” All of the tissue sections were acquired after setting the homogenous Köhler illumination, and white balance was also set before saving the lamp intensity and exposure time. Moreover, all of the tissue sections were acquired using the same acquisition profile. Based on the properties of the original tissue, different threshold level intensities were identified in technical control tissue sections stained with mouse monoclonal isotype control primary antibody. Therefore, cut-off values of 7–11 were used to define xCT-positive cells.

### 2.4. TCGA-HNSCC Dataset Analysis

The publicly available Cancer Genome Atlas (TCGA) dataset was analyzed via the Kaplan–Meier (KM) plotter portal (https://kmplot.com/analysis/ accessed on 1 June 2022). From the HNSCC-TCGA repository data from 500 patients with available RNA-seq, HTSeq counts, and survival information were processed. Detailed information regarding the TCGA data structures and methods used can be reviewed on the TCGA website (https://tcga-data.nci.nih.gov/tcga/ accessed on 15 June 2022). Analyses and visualizations were performed according to the KM plotter guidelines and protocols [37,38] The patient samples were split into two cohorts according to the low or high levels of xCT gene expression. The two subgroups were compared using a KM survival plot, and the hazard ratio with 95% confidence intervals and the log-rank *p* value were calculated.

### 2.5. Cell Lines and Cell Culture

The head and neck squamous cell carcinoma (HNSCC) cell lines FaDu and SCC25 were purchased from the American Type Culture Collection (ATCC) (Wesel, Germany). They were cultured in a humidified atmosphere at 37 °C under 5% CO_2_. FaDu cells were grown in minimum essential medium Eagle (Sigma-Aldrich, St. Louis, MI, USA). The culture medium was supplemented with 10% fetal bovine serum (FBS) (HyClone™) (Thermo Fisher Scientific, Vienna, Austria), 1% penicillin/streptomycin/glutamine (100×) (PSG) (Gibco, Thermo Fisher Scientific, Waltham, MA, USA), and 0.1 mM non-essential amino acids (Gibco, Life Technologies, Grand Island, NY, USA). SCC25 cells were maintained in DMEM/F12 (1:1) medium (Gibco, Thermo Fisher Scientific, Waltham, MA, USA) supplemented with 10% FBS, 1% penicillin/streptomycin/glutamine and 400 ng/mL Hydrocortisone (Sigma-Aldrich, St. Louis, MI, USA).

### 2.6. Cell Viability Assay

To evaluate cell viability, HNSCC cells were seeded in 6-well plates (1 × 10^5^ cells/well) and allowed to recover and adhere overnight. Cells were treated with erastin (2 μM), ferrostatin-1 (1 µM), ravoxertinib (100 nM), or their combination for the appointed time points. As a control, the cells were treated with dimethylsulfoxide (DMSO; <0.01%). After the treatment period, cell viability was evaluated using a test based on trypan blue dye exclusion as previously described [39,40]. Briefly, cells were trypsinized and counted using a Beckman Coulter Vi-CELL AS cell viability analyzer (Beckman Coulter, Fullerton, CA, USA). The number of viable cells and total cell number were determined in the control and treated HNSCC cell samples. The cell viability was expressed as the percentage of viable cells relative to the total cells.

### 2.7. Three-Dimensional Tomographic Microscopy

FaDu and SCC25 cells were seeded into the ibiTreat polymer coverslip µ-dishes with a diameter of 35 mm and high walls (ibidi GmbH, Graefelfing, Germany). After overnight incubation, cells were treated with erastin (2 µM) and ferrostatin-1 (1 µM) and kept at 37 °C under a 5% CO_2_ humidified atmosphere. At the time point of 48h after treatment, cells were analyzed to determine their morphology and the characteristics of ferroptotic cell death development using a 3D tomographic microscope with a 60× objective—3D-Explorer-FLUO (Nanolive SA, Tolochenaz, Switzerland)—and STEVE software (Nanolive SA, Tolochenaz, Switzerland).

### 2.8. Lipid Peroxidation

Untreated and treated cells were incubated with 2.5 μM of C11-BODIPY581/591 (Life Technologies, Carlsbad, CA, USA) for 30 min following the manufacturer’s instructions and subjected to flow cytometry. Flow cytometric analyses were performed on a BD FACSCanto™ II Flow Cytometry System (Becton, Dickinson and Company, Franklin Lakes, NJ, USA) using BD FACSDiva™ Software 7.0 (Becton, Dickinson, and Company, Franklin Lakes, NJ, USA) for acquisition, and further analysis was performed with FlowJo v10 software (Becton, Dickinson & Company). A minimum of 10,000 cells were analyzed in each condition.

### 2.9. Measurement of ROS Levels

A DCFDA/H2DCFDA—Cellular ROS Assay Kit (catalog No. ab113851; Abcam, Cambridge, MA, USA) was used according to the manufacturer’s instructions to quantitatively assess reactive oxygen species in live HNSCC cell samples. The ROS formation in untreated and treated HNSCC cells was detected by utilizing the cell-permeable reagent 2′,7′-dichlorofluorescein (DCFDA, also known as H2DCFDA, DCFH DA, and DCFH), which ROS oxidized to form a fluorescent compound with excitation and emission spectra of 495 and 529 nm, respectively.

### 2.10. Analysis of Erk1/2 Activation

For the detection of phosphorylated and total ERK1/2 proteins, the Erk1 (phospho T202 + Y204) + Erk2 (phospho T185 + Y187) + Total ELISA Kit (ab176660; Abcam, UK) was used. Cells were treated with ravoxertinib, erastin, or their combination as described above, and then the assay was processed following the manufacturer’s guidelines. Each blank well was loaded with 100 μL of sample diluent, while the residual wells were loaded with either 100 μL of standard or sample diluent for the test. The optical density (OD) value of each well at 450 nm was determined using a micro-plate reader.

### 2.11. Promega Lumit™ Immunoassay Cellular System

The Lumit™ Immunoassay Cellular System, a homogeneous bioluminescent assay, was used to measure the superoxide dismutase (SOD) protein levels in cell lysates in the untreated control and treated cell samples. This assay combines immunodetection and NanoLuc^®^ Binary Technology (NanoBiT^®^). Briefly, Large BiT (LgBiT) and Small BiT (SmBiT) subunits were conjugated to a pair of secondary antibodies against two different species (anti-mouse and anti-rabbit). Seeded cells were lysed in 96-well plates using a NanoBiT compatible lysis solution (Digitonin), and the target protein was detected by adding an antibody mix containing two primary antibodies against the target protein along with SmBiT- and LgBiT-conjugated secondary antibodies. Appropriate primary antibody pairs were used: SOD1 (E4G1H) XP^®^ Rabbit mAb #37385 and SOD1 (71G8) Mouse mAb #4266; SOD2 (D3X8F) XP^®^ Rabbit mAb #13141 and SOD2 Monoclonal Antibody (GT582) Catalog # MA5-31514 Invitrogen (Mouse). The binding of the primary/Lumit secondary antibody complexes to their corresponding epitopes meant the NanoBiT subunits were brought into proximity to form an active NanoLuc luciferase that generated light in proportion to the amount of target protein. Luminescence data were normalized to the cell number per well by adding GF-AFC Substrate to all the cells 40 min before lysis. The graphs were plotted with luminescence RLU (Relative Light Unit) values normalized to fluorescence RFU (Relative Fluorescence Unit) values.

### 2.12. Statistical Analysis

GraphPad Prism version 8.0 software (GraphPad Software, San Diego, CA, USA) was used to blot the graphs and for statistical evaluation. All the values are represented as the means ± standard error of the mean or standard deviation. If not stated otherwise, the differences between samples were analyzed with the Student’s t-test. The results regarding the (∆CT) relative gene expression of SLC7A11 in the control normal mucosa samples showed normal Gaussian distribution, whereas the relative gene expression of SLC7A11 was not normally distributed. In this case, the Mann–Whitney test was required to compare the median of the control and HNSCC samples. The statistical significance of the *p*-value is designated with an asterisk (*); *p*-values: * *p* < 0.05, ** *p* < 0.01, *** *p* < 0.001.

## 3. Results

### 3.1. xCT (SLC7A11) Is Overexpressed in Malignant Tissues and Is Negatively Associated with the Overall Survival Rate in HNSCC Patients

In order to better understand whether xCT (SLC7A11) can serve as a potential therapeutic target in patients with HNSCC, we analyzed a correlation between the level of xCT (SLC7A11) expression and the overall survival rate in patients with HNSCC using the TCGA data (Figure 1a). Thus, it was observed that patients with HNSCC with high xCT expression demonstrated a significantly (*p* = 0.0038) lower overall survival rate compared to the cohort of patients with low xCT expression. Thus, the cohort with low xCT expression displayed the median survival time of 65.73 months, whereas patients with high xCT expression displayed a median survival time of 32.93 months.

Additionally, we investigated the mRNA levels in 10 normal epithelium and 37 tumor samples obtained from patients with HNSCC (Figure 1b). Malignant tissues showed ~4.3-fold higher xCT (SLC7A11) gene expression compared to normal mucosa (*p* = 0.0214). Next, xCT (SLC7A11) protein expression was evaluated in normal epithelium and carcinoma tissues (Figure 2). It was demonstrated that non-malignant tissues had low xCT-positive-cell contents (9.39 + 5.51%) (Figure 2a,d), whereas the tumor samples contained a significantly higher number of xCT-positive cells (49.93 + 7.03%) (Figure 2b–d). The xCT immunohistochemical reaction was present in the cancer cell nests and not in stroma cells. These data revealed the overexpression of xCT in HNSCC tissue against normal mucosa and, moreover, a tumor-cell-specific, mainly membranous localization.

### 3.2. xCT (SLC7A11) Overexpressing HNSCC Cell Lines FaDu and SCC25 Demonstrate Sensitivity to Ferroptosis Inducer Erastin

First, FaDu and SCC25 HNSCC cells were analyzed for xCT (SLC7A11) expression. It was found that both of the cell lines investigated expressed xCT at high levels with ~48% and ~95% positive cells in the FaDu and SCC25 cell lines, respectively (Figure 3a,b).

Next, we evaluated whether FaDu and SCC25 carcinoma cells are susceptible to erastin treatment. After the preliminary experiments using different doses of erastin, the IC50 values for both cell lines were determined. It was found that the IC50 values for FaDu and SCC25 cells at 72 h were 1.4 µM and 2.1 µM, respectively. It was decided to use erastin at a dose of 2 µM in all of the experiments performed. After the administration of erastin (2 µM), the FaDu and SCC25 cells demonstrated a significant reduction in cell viability (34.23 + 3.95% for FaDu cells and 50.49 + 1.99% for SCC25 cells) (Figure 4a).

As ferroptosis is a type of cell death caused by iron-dependent lipid peroxidation [41], we also determined whether cell viability would be affected after HNSCC cell treatment with ferrostatin-1, a potent lipophilic antioxidant. Indeed, ferrostatin-1 fully abrogated the decrease in cell viability caused by erastin in both FaDu and SCC25 carcinoma cells (Figure 4a).

It was recently found that ferroptotic cell death is characterized by specific morphological changes which are not observed during other types of cell death [42,43]. Indeed, erastin treatment induced the formation of a ballooning phenotype, representing clear rounded cells with empty cytosol (Figure 4b). It is interesting to note that ferrostatin-1 abolished erastin-induced, ferroptosis-specific morphological alterations in carcinoma cells, and the HNSCC cells investigated did not differ from the untreated controls.

### 3.3. HNSCC Cells Are Characterized by Expression of Phosphorylated Erk1/2

As the activation of Erk1/2 intracellular signaling can be associated with the affected cell’s response to cytotoxic agents, we then determined how the total and phosphorylated Erk1/2 (pT202/Y204; T185/Y187) forms are balanced in FaDu and SCC25 carcinoma cells before and after erastin treatment. Although the constitutive total Erk1/2 expression was ~2.6-fold higher in FaDu than in SCC25 cells, the phosphorylation levels were comparable in both cell lines. The Erk inhibitor ravoxertinib (100 nM) did not markedly change the total Erk1/2 expression in both of the carcinoma cell lines (Figure 5a). However, the phosphorylation of Erk1/2 was significantly inhibited by ravoxertinib in these HNSCC cells (Figure 5b). Erastin alone or in combination with ravoxertinib demonstrated the same levels of the total and phosphorylated Erk1/2, as was observed in the untreated cells.

### 3.4. Erk1/2 Inhibitor Ravoxertinib Mitigates HNSCC Cell Susceptibility to Erastin

To determine whether Erk1/2 phosphorylation plays a role in HNSCC cells’ susceptibility to erastin, we evaluated the cell viabilities in response to ravoxertinib, erastin, and their combination (Figure 6a,b). Ravoxertinib 100 nM alone did not show any pronounced effect on the HNSCC cells at 24, 48, and 72 h after treatment, whereas 2 µM of erastin alone demonstrated a significant reduction in cell viability in the FaDu and SCC25 cells. As it is seen, the FaDu cells were ~1.99-fold more sensitive to erastin than the SCC25 cells, and the cell viability in the FaDu cells was 27.86 + 5.96% versus 55.51 + 4.97% in the SCC25 cells 72 h after erastin administration. The HNSCC cells pretreated with ravoxertinib demonstrated increased cell viability after erastin treatment compared to erastin alone. Thus, after 72 h of HNSCC cell incubation with ravoxertinib and erastin, the cell viability in the FaDu and SCC25 cells was 44.25 + 4.04% and 70.05 + 4.39% versus 27.86 + 5.96% and 55.51 + 4.97% after treatment with erastin alone, respectively.

Next, the level of lipid peroxidation was studied in HNSCC cells after their treatment with ravoxertinib, erastin, or their combination (Figure 6c). It was found that ravoxertinib alone did not promote lipid peroxidation in carcinoma cells. Erastin alone effectively increased (~6.41-fold for FaDu cells and ~6.66-fold for SCC25 cells) lipid peroxidation in HNSCC cells compared to control cells, and ravoxertinib protected the cells from the erastin-induced peroxidation of the lipids and reduced the peroxidation by ~1.8-fold in FaDu cells and ~1.6-fold in SCC25 cells compared to erastin treatment.

### 3.5. Erk1/2 Inhibitor Ravoxertinib Decreases ROS Production and Enhances Expression of ROS Scavengers in HNSCC Cells in Response to Erastin

As lipid peroxidation is closely related to reactive oxygen species (ROS) production, we next determined whether ravoxertinib, erastin, or their combination can modulate the levels of total ROS in HNSCC cells (Figure 7a). Ravoxertinib alone did not affect ROS production in the cells in comparison with the control untreated cells. In contrast, erastin alone significantly increased the ROS levels in both the FaDu and SCC25 cells (~6.6-fold in the FaDu cells and ~7.5-fold in the SCC25 cells) if compared with the untreated control. Similar to lipid peroxidation, cell pretreatment with ravoxertinib diminished the ROS production in erastin-treated HNSCC cells. In the FaDu and SCC25 cells, after treatment with a combination of ravoxertinib and erastin, the MFIs were 669.9 + 82.39 and 416.8 + 189.0 versus 1163.0 + 167.8 and 848.1 + 109.1 in erastin-treated cells, respectively.

It is known that ROS levels can be regulated by ROS scavengers [44,45]. Superoxide dismutases 1 and 2 (SOD1 and SOD2) are involved in the modulation of Erk1/2-associated ROS production [46,47,48]. Hence, it was decided to evaluate the treatment-dependent SOD1 and SOD2 expressions in HNSCC cells. Constitutive levels of SOD1 and SOD2 were ~1.82- and ~1.92-fold higher in the FaDu than in the SCC25 cells, respectively (Figure 7b,c). Additionally, the FaDu cells contained ~2-fold more active SOD1 and SOD2 expressions in response to ravoxertinib compared to the SCC25 cells. Interestingly, erastin did not markedly change the SOD1 and SOD2 expressions in both cell lines, and the combination of ravoxertinib and erastin induced SOD1 and SOD2 expressions at levels comparable with Erk1/2 inhibitor-caused expressions. There were no significant differences in the SOD1 and SOD2 expression in response to the combination treatment compared to the ravoxertinib treatment alone in both the FaDu and SCC25 carcinoma cells.

## 4. Discussion

HNSCC is a malignant tumor characterized by unfavorable clinical outcomes despite the use of novel and promising treatment approaches [49]. Therefore, new and more effective therapeutic strategies are needed to improve the disease-free and overall survival rates in patients with HNSCC. The treatment efficacy can be reduced if a malignant tumor possesses an activation of the molecular pathways associated with the downregulation of cell death programming. Luckily, different cell death types are not similarly and concomitantly repressed in carcinoma cells, and the inhibition of a selected cell death pathway does not necessarily lead to cancer cell insensitivity in another kind of cell killing mode. Therefore, the key molecular target which regulates the most effective pathway to kill the malignant cells could be identified.

It is generally accepted that a promising therapeutic target should be overexpressed and functionally essential in malignant tumors and non-essential for normal cell physiology; this targeted molecule should also have a causative link with tumorigenesis and, finally, it should be druggable. Recently, a new type of cell death was discovered and named ferroptosis [5,6]. It was shown that the xCT (SCL7A11) molecule is a key negative regulator of ferroptosis. It is known that the xCT (SLC7A11) molecule is involved in tumor growth, chemo- and radiotherapy resistance, and poor clinical outcomes [50,51,52,53]. Hence, xCT was considered as a potential and promising target to treat cancer [53]. Although recent scientific reports clearly demonstrated that ferroptosis can be effectively activated in HNSCC cells, little is known about the molecular regulation of HNSCC cell sensitivity and susceptibility to ferroptosis.

In order to better understand a role of xCT (SLC7A11) in HNSCCs, we first evaluated whether this molecule is differently expressed in malignant and normal tissues in patients with HNSCC. As cancerous tissue was shown to demonstrate an overexpression of xCT, and normal mucosa contained a limited number of the xCT-positive cells, it seems that the xCT molecule could be suggested as a potential target in patients with HNSCC. Furthermore, our analysis of a correlation of xCT expression and overall survival also confirmed a role of this molecule in unfavorable prognoses in patients with HNSCC. These data are in agreement with the previously published reports which showed a shortened disease-free and overall survival time in patients with cancer associated with xCT overexpression [54,55]. Assuming that xCT can be targeted to kill carcinoma cells, we performed these experiments with erastin, an xCT inhibitor [56]. Although both cell lines were characterized by the overexpression of xCT, the FaDu carcinoma cells were more susceptible to erastin-caused ferroptosis induction than the SCC25 cells. This fact allows us to hypothesize that either the level of xCT expression or other molecular patterns of carcinoma cells can modulate cell sensitivity to erastin. Our own data demonstrate that the level of xCT expression does not correlate with cellular responses to erastin treatment. Hence, we presume that cell sensitivity to erastin can be dependent on any specific molecular profiling of carcinoma cells. Considering that erastin-caused ferroptosis is not still fully understood on the molecular level, the continuous and comprehensive investigation of this mode of cell death should be carried out.

Ferroptosis was first described in 2012, and it was shown that this type of cell death was caused by iron-induced lipid peroxidation [5]. In order to achieve an optimal induction of lipid peroxidation, detoxifying intracellular machinery should be downregulated [57]. Currently, a number of molecules have been described to be implicated in the suppression of ferroptosis development, such as GSH, GPX4, and Nrf2 [10,11,16,58]. However, later experimental research allowed us to observe that lipid peroxidation can be triggered by other stimuli [59]. Thus, either constitutive or treatment-induced oxidative stress can lead to ferroptosis development [60]. The crosstalk between oxidative stress and the EGFR-related pathway is an important regulatory mechanism in carcinogenesis, tumor progression, and therapy resistance [61]. The canonical EGFR pathway accompanied by the activation of MAP kinases; mainly, Erk1/2, is a driver of HNSCC development and progression [27,30].

As oxidative stress can be accompanied by the activation of MAP kinases, affecting both pro-survival and pro-death signaling in carcinoma cells [62], we have the expression of the Erk1/2 protein in FaDu and SCC25 HNSCC cells. It was observed that the inhibition of Erk1/2 phosphorylation in HNSCC cells through the use of a non-toxic concentration of the Erk inhibitor, ravoxertinib, resulted in the enhancement of the erastin-dependent increase in FaDu and SCC25 cell viability. Therefore, it is possible to suppose that carcinoma cells possessing an expression of phosphorylated Erk1/2 are more susceptible to ferroptosis induction. Very similar data were recently presented in relation to erastin-treated pancreatic cancer cells [63]. It was shown that pancreatic cell sensitivity to ferroptosis can be realized via Erk1/2 and Jnk signaling. Additionally, we clearly demonstrated that phosphorylated Erk1/2 plays a role in the erastin-induced lipid peroxidation. Thus, the ravoxertinib-dependent inhibition of Erk1/2 phosphorylation led to the protection of the cells from the erastin-induced lipid peroxidation. Although the role of Erk1/2 in ferroptosis induction has already been discussed in a number of studies [24,64,65], the role was not elucidated with regard to HNSCC cells, and our research may result in the development of additional biomarkers to predict HNSCC cells’ response to erastin.

In our study, the constitutive ROS levels were not very pronounced in both the untreated FaDu and SCC25 cells, and erastin significantly elevated the production of intracellular ROS. We did not observe any concomitant upregulation of Erk1/2 expression in response to erastin, but the Erk inhibitor, ravoxertinib, downregulated ROS levels and lipid peroxidation in HNSCC cells. It is known that the most prevalent lipids that affect ROS are hydroxyl radicals [62]. Therefore, it is logical to assume that the radical scavengers SOD1 and SOD2 can be implicated in the regulation of the erastin-induced decrease in the viability of HNSCC cells. Indeed, SOD1 and SOD2 are the key antioxidant enzymes which regulate oxidative stress in cells [66], and they prevent the formation of hydroxyl radicals [67,68]. Subburayan et al. recently reported that superoxides play a key role in the induction and development of ferroptosis triggered by cysteine starvation [69]. Furthermore, SOD1 and SOD2 are involved in the regulation of Erk1/2-dependent ROS levels [46,47,70,71]. A strong relationship between Erk1/2 and SOD1 and SOD2 expression was demonstrated in our study. Thus, we observed that the inhibition of Erk1/2 phosphorylation was accompanied by the upregulation of the antioxidants SOD1 and SOD2 in HNSCC cells. Additionally, the increased expression of these ROS scavengers upon combination treatment using ravoxertinib and erastin resulted in improved cell viability compared to the cell treatment with erastin alone (Figure 8).

Therefore, it is possible to conclude that the inhibition of xCT caused by erastin in carcinoma cells seems to be a promising approach and can be further developed as an additional strategy for HNSCC treatment. As ferroptosis induction via erastin is strongly dependent on the expression of Erk1/2 associated with the modulation of the ratio between ROS production and expression of ROS scavengers, phosphorylated Erk1/2 can be used as a predictor for cancer cells’ responses to erastin.

## Figures and Tables

**Figure 1 cells-12-00336-f001:**
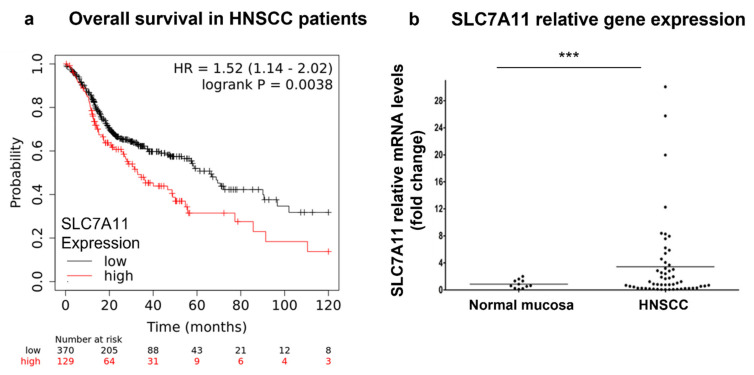
xCT (SLC7A11) expression in patients with HNSCC. The prognostic value of xCT (SLC7A11) mRNA was evaluated in patients with HNSCC using the KMplot database (https://kmplot.com/analysis/), and correlation between the levels of xCT (SLC7A11) and overall survival rate is displayed on the plot diagram (**a**). Relative xCT (SLC7A11) mRNA levels in tissue specimens of patients with HNSCC and control healthy mucosa were determined using real-time PCR analysis (**b**). Relative quantities of the transcripts were calculated via pairwise differences of threshold cycles (∆CT) of the SLC7A11 gene and the loading control GAPDH housekeeping gene. *** *p* < 0.001.

**Figure 2 cells-12-00336-f002:**
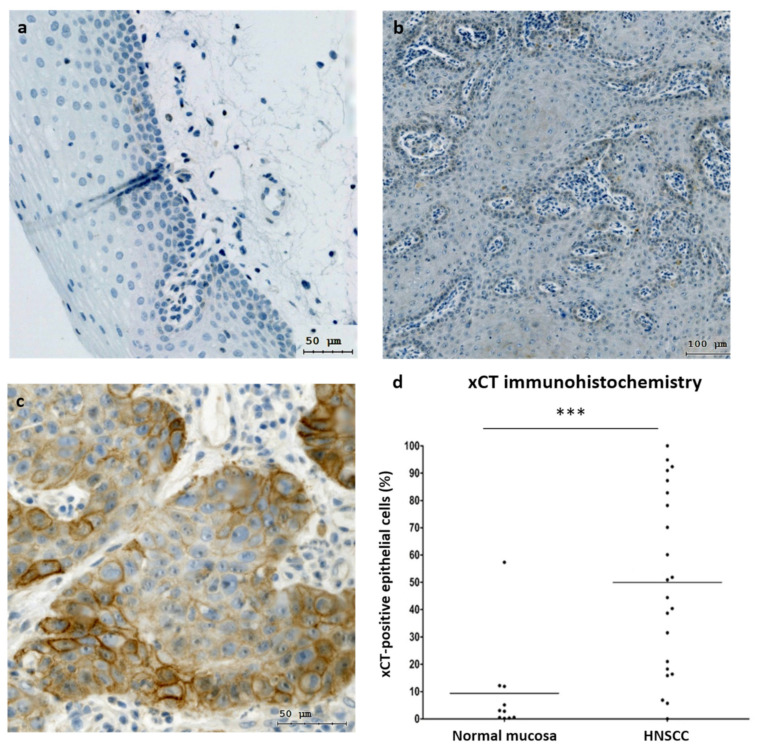
xCT (SLC7A11) expression in normal mucosa and HNSCC samples. Immunohistochemical representation of xCT obtained via TissueFax® (TissueGnostocs™, Vienna, Austria) in normal mucosa (**a**) and tumor tissue (**b**,**c**); 400× magnification, bar: 50 µm (**a**,**c**); 200× magnification, bar: 100 µm (**b**). Differences in xCT expression in normal mucosa and HNSCC samples (**d**). Statistical analysis was performed with the Mann–Whitney test, *** *p* < 0.001.

**Figure 3 cells-12-00336-f003:**
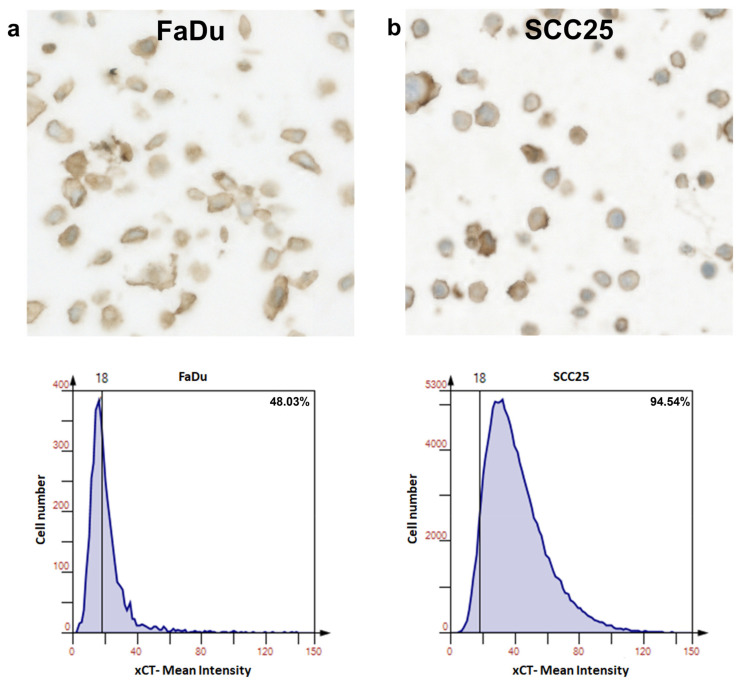
Expression of xCT (SLC7A11) in FaDu and SCC25 HNSCC cells. Representative immunohistochemical images made using TissueFax^®^ (TissueGnostocs™, Vienna, Austria) show the levels of the enzyme content in FaDu (**a**) and SCC25 (**b**) cells. Mean intensity was determined using the HistoQuest software.

**Figure 4 cells-12-00336-f004:**
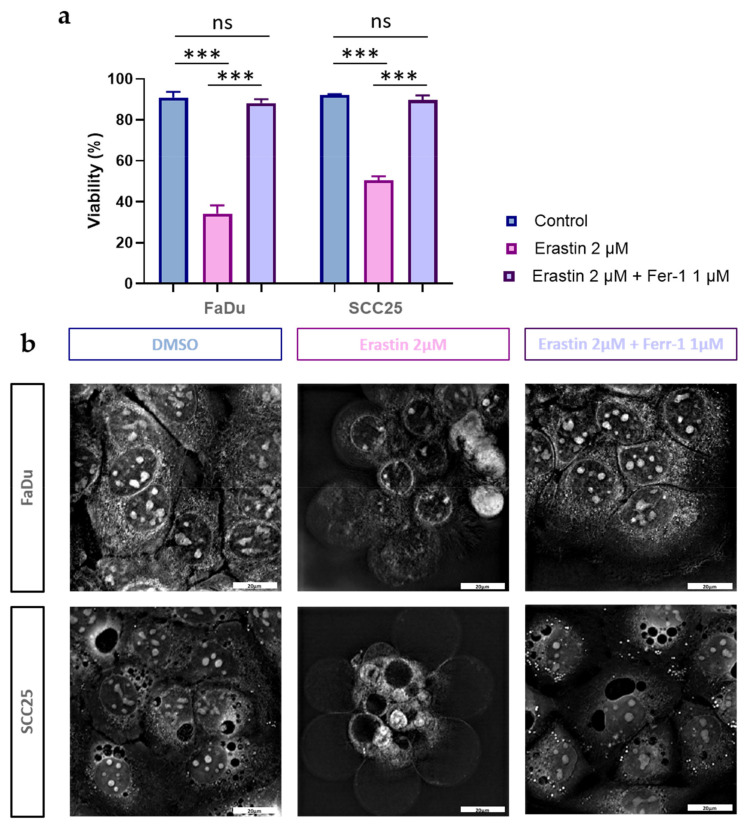
Effects of erastin and ferrostatin-1 on viability and morphology of HNSCC cells. Cell viability was determined in FaDu and SCC25 cells at 72 h after treatment with erastin (2 µM) or combination of erastin (2 µM) and ferrostatin-1 (1 µM). DMSO was used as a vehicle control. Columns represent the mean + SEM calculated after at least three independent experiments, *** *p* < 0.001 (**a**,**b**) Morphological characteristics of ferroptotic cell death in HNSCC FaDu and SCC25 cells were evaluated using holotomographic microscopy (Nanolive SA 3D-Explorer-FLUO, Tolochenaz, Switzerland) as described in Materials and Methods. Cells were treated with erastin (2 µM) or a combination of erastin (2 µM) and ferrostatin-1 (1 µM), and at 72 h after treatment, the representative images were obtained; 60× magnification, bar: 20 µm.

**Figure 5 cells-12-00336-f005:**
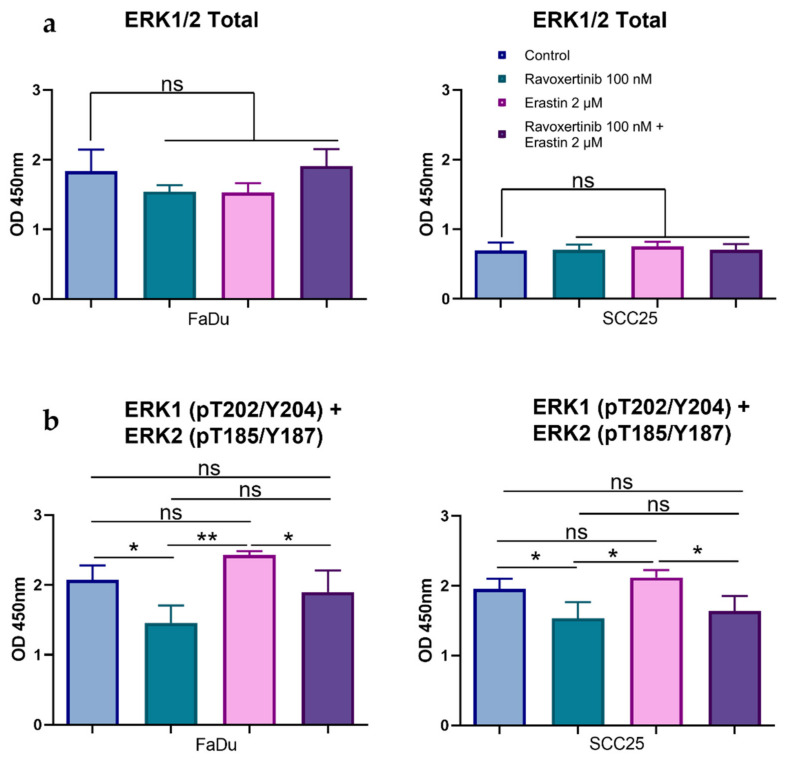
Effects of erastin and ravoxertinib on the expression of total and phospho-Erk1/2. Levels of the total Erk1/2 (**a**) and phospho-Erk1 (T202 + Y204) and –Erk2 (T185 + Y187) (**b**) were determined in FaDu and SCC25 cells 24 h after treatment with erastin (2 µM), ravoxertinib (100 nM), or their combination. Data are presented as mean + SD after three independent experiments, * *p* < 0.05, ** *p* < 0.01.

**Figure 6 cells-12-00336-f006:**
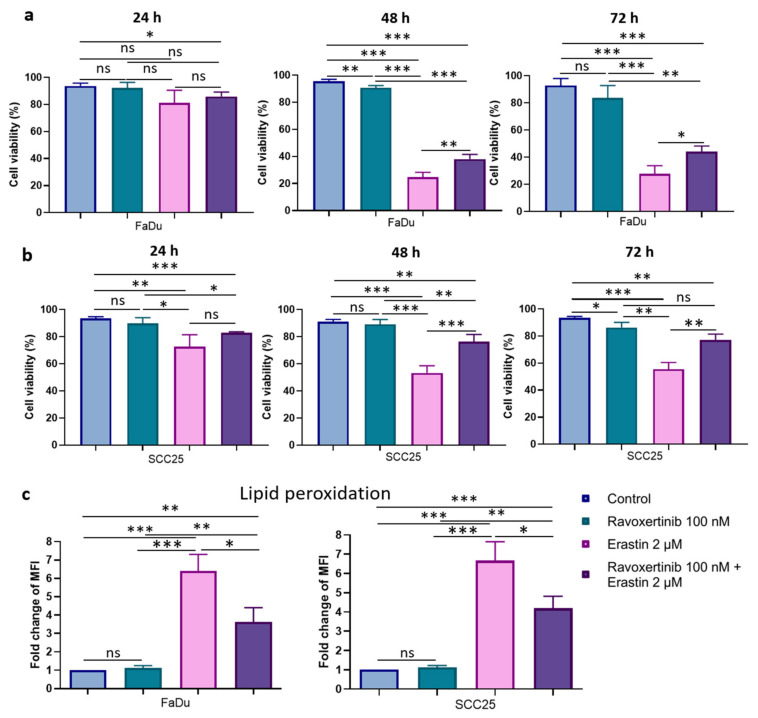
Cell viability and lipid peroxidation in FaDu and SCC25 HNSCC cells in response to erastin and ravoxertinib exposure. FaDu (**a**) and SCC25 (**b**) cells were treated with erastin (2 µM), ravoxertinib (100 nM), or their combination, and cell viability was detected at 24, 48, and 72 h, as described in Materials and Methods. Lipid peroxidation was studied via flow cytometry analysis of the oxidized C11-BODIPY (581/591) in HNSCC cells at 24 h after treatment with erastin (2 µM), ravoxertinib (100 nM), or their combination (**c**). Data are given as mean + SD (**a**,**b**) or mean + SEM (**c**) obtained from at least three independent experiments, * *p* < 0.05, ** *p* < 0.01, *** *p* < 0.001.

**Figure 7 cells-12-00336-f007:**
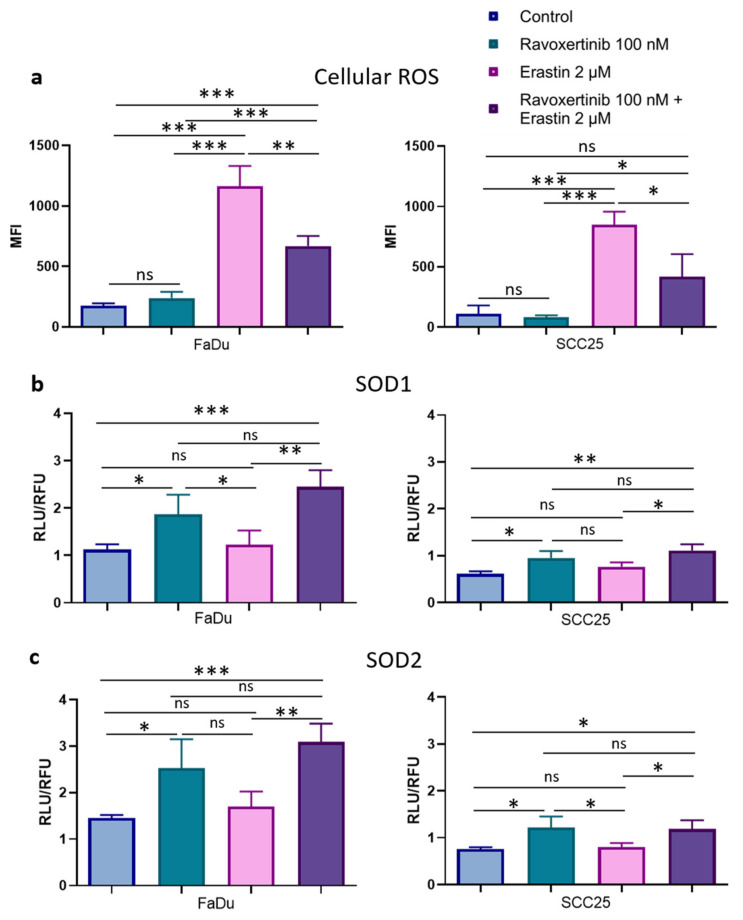
ROS production and expression of ROS scavengers, SOD1, and SOD2 in HNSCC cells. Flow cytometry analysis of the HNSCC cells stained with DCFDA fluorogenic dye was used to measure hydroxyl, peroxyl, and other ROS activities in FaDu and SCC25 cells 24 h after treatment with erastin (2 µM), ravoxertinib (100 nM), or their combination (**a**). Expression levels of ROS scavengers SOD1 (**b**) and SOD2 (**c**) were evaluated using the Promega Lumit™ Immunoassay Cellular System (see Materials and Methods) 24 h after treatment with erastin (2 µM), ravoxertinib (100 nM), or their combination. Data are given as mean + SD (**a**) or mean + SEM (**b**,**c**) obtained from at least three independent experiments, * *p* < 0.05, ** *p* < 0.01, *** *p* < 0.001.

**Figure 8 cells-12-00336-f008:**
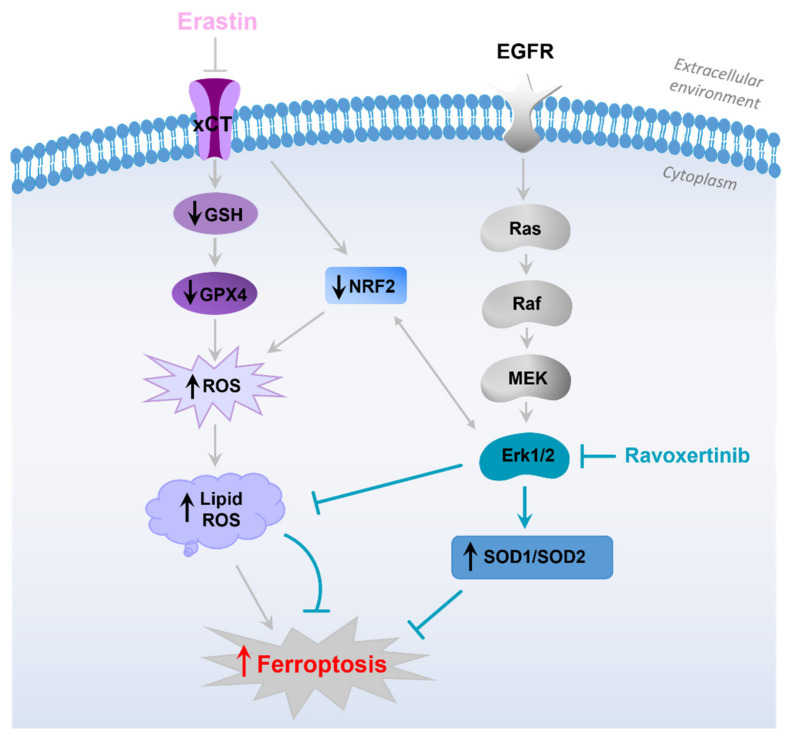
Hypothetical sequence of events leading to development of ferroptosis in HNSCC cells in response to erastin treatment. The schematic diagram illustrates that erastin inhibits xCT, resulting in downregulation of GSH, GPX4, and Nrf2 [58]. Decrease in the detoxifying molecules caused by erastin leads to enhanced ROS production, lipid peroxidation, and ferroptosis development [19,57]. EGFR pathway and its downstream Erk1/2 activation plays an important role in HNSCC pathophysiology. Erk1/2, in cooperation with Nrf2 [24], is involved in the modulation of ROS formation and expression of ROS scavengers SOD1 and SOD2. Inhibition of Erk1/2 signaling via specific Erk-inhibitor, ravoxertinib, affects the ratio between ROS levels and ROS scavengers’ expression, resulting in ferroptosis attenuation.

## Data Availability

All data supporting reported results are contained within the article.

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
