# Peer review of "Erk1/2-Dependent HNSCC Cell Susceptibility to Erastin-Induced Ferroptosis"

_cells, 2023, doi:10.3390/cells12020336_

Round 1
Reviewer 1 Report
The manuscript by Dr. Skvortsova and the group elaborates on the mechanism of erastin-dependent activation of ferroptosis. This promising approach can be further developed as a novel strategy for HNSCC treatment. This is a hypothesis-driven research paper that follows a therapeutic implication. It is a well-written manuscript, though a few things need to be addressed before it is ready for acceptance. They are as follows:
1. It has recently been shown that mutant KRAS activates NRF2 (PMID: 21734707) and glutamine limitation induces pro-ferroptosis stimuli, including inhibition of GPX4, in KRAS-mutant pancreatic cancer cells, suggesting that RAS-mutant cancer cells displaying high levels of glutaminolysis might be more susceptible to ferroptosis (PMID: 31911550, PMID: 33870211). This is an important aspect that authors must discuss in a few lines in the introduction part.
2. Authors should add a model based on this manuscript's published literature and findings. This model can be added as the last figure and be discussed in the discussion part.
3. Lastly, another important aspect authors must discuss that how NRF2 pathways play a significant role in HNSCC. Since it has been shown that the Over-activation of NRF2 promotes the malignant progression of HNSCC through reprogramming G6PD- and TKT-mediated nucleotide biosynthesis. Targeting NRF2-directed cellular metabolism is an effective strategy for developing novel treatments for head and neck cancer (PMID: 33859744). Furthermore, it has also been shown that Nrf2 attenuates ferroptosis-mediated IIR-ALI by modulating TERT and SLC7A11 (PMID: 34716298). The authors need to add a few lines discussing this point in the introduction.
Author Response
- Thank you very much for your very important comment. Indeed, ferroptosis is a complex process involving a number of the regulatory molecules, and GPX4 and Nrf2 are the key regulators of ferroptosis. The manuscript has been updated and the section „Introduction“ now contains additional information about the mentioned signaling pathways. The mentioned references are also included tot he list of the referenced articles.
- Figure 8 has been added in the section „Discussion“
- Now the section „Introduction“ contains information about the role of Nrf2 in ferroptosis development, and mentioned articles are referenced in the manuscript.
Reviewer 2 Report
The current work is well written and the data are of great interest for the readership. Therefore this work should be published in its present form.
In their study, Savic D et al. investigated if Erk1/2-dependent head and neck squamous cell carcinoma (HNSCC) cells are vulnerable to ferroptosis induction.
This topic has some merit because ferroptosis is a new field of iron metabolism and the study design is original.
The authors showed that ravoxertinib-dependent inhibition of Erk1/2 signaling led to the decrease of erastin efficacy due to the affected ROS production. The manuscript is well written and the conclusion concise. Therefore, this study should be published in its present form.
Author Response
Thank you very much for your very positive evaluation of the manuscript.
Reviewer 3 Report
The study examined xCT expression in head and neck squamous cell carcinoma and its possible signaling pathway to ferroptosis induction. It was found that xCT was overexpressed in HNSCC, which was likely to increase the viability of HNSCC cells by suppressing ferroptosis. Erk phosphorylation was found responsible for the erastin-induced cell death by upregulation of ROS production. Overall, the study presented a possible connection between xCT overexpression and cell viability via Erk signaling in HNSCC. However, the following points need to be considered.
1. The introduction section should include the current status of erastin treatment in HNSCC.
2. Each subtitle in the result section should reflect the key findings in that subsection, rather than what study was performed.
3. The rationale connecting one experiment to another should be discussed more unbiasedly. For instance, why was Erk chosen to examine, not other MAPK pathways such as JNK or p38, which are more common regulators of oxidative stress?
4. The language throughout the paper needs to be tightened by using more straightforward words. For instance, the introduction section should present the background and the logic for the study, while the discussion section should contrast your findings against the background and make them stand out.
Author Response
Authors are grateful for valuable comments of the reviewer 3.
- Information about the efficacy of erastin to kill HNSCC cells is provided in the section „Introduction“ (References 16-18)
-
Thank you very much for your comment. We have modified all the subtitles and they now describe the key findings.
-
The rationale to investigate an activation of Erk1/2 has been provided in the sections „Introduction“ and „Discussion“
-
We have modified the sections „Introduction and „Discussion“, and the English Editor of the MDPI has edited the text of the manuscript. Authors hope that the reviewer will find that the manuscript has been improved due to the helpful comments of the reviewer.
Round 2
Reviewer 1 Report
All concerns have been addressed- ready for acceptance.
Author Response
Many thanks to this reviewer for the positive evaluation of our manuscript.
Reviewer 3 Report
The authors have revised their manuscript moderately. However, several issues remain with the study.
1. For the result section 3.1, the subtitle “xCT is overexpressed in malignant tissues and is associated with reduced overall survival” is unclear. Based on the study, xCT supports cancer cell survival. Therefore, the authors need to clarify by saying “…..and is negatively associated with the overall survival rate of the patients”.
2. For the result section 3.2, normal head/neck cell lines are needed as controls to compare with the two cancer cell lines in order to determine whether xCT was indeed overexpressed in the cancer cells. And also, what does erastin do to the normal cells?
3. Again, for the result section 3.3, “HNSCC cells are characterized by expression of phosphorylated Erk1/2”, normal cell lines are needed for comparison. Otherwise, what is the point to examine Erk phosphorylation? It can happen to any cell.
4. The study started with xCT overexpression in cancer and then switched to ferroptosis induction, while the connection between these two is weak. In order to smoothen the logic of the study, the authors need to knock down xCT and examine how it affects ferroptosis, Erk phosphorylation, erastin treatment, and so on.
Author Response
- Many thanks for this valuable comment. We have changed the mentioned subtitle
-
Authors thank the reviewer for his/her opinion about the evaluation of xCT expression in non-malignant cells. However, we have to emphasize that the lower expression of the xCT in normal mucosa was demonstrated on the gene and protein levels using the patients‘ samples. Authors suppose that this is a sufficient confirmation of the differences in xCT expression in non-malignant and malignant tissues.
As malignant tissues were characterized by overexpression of the xCT, we further made all the experiments in xCT overexpressing HNSCC cells. Erastin is a specific inhibitor of xCT, and we did not aim to investigate the response to erastin in non-malignant cells without expression of the target for erastin.
Unfortunately, our lab does not possess any mucosal cell lines, and we are unable to perform any experiments with non-malignant cells.
-
The reason to evaluate Erk1/2 phosphorylation in the investigated HNSCC cells is described in the section „Introduction“ and additionally explained in the section „Discussion“. Thus, more than 90% of HNSCCs demonstrate EGFR-related activation of Erk1/2 in tumor tissues. Therefore, it was decided to evaluate a role of the Erk1/2 in erastin-caused ferroprosis in HNSCC cells.
Unfortunately, our lab does not possess any mucosal cell lines, and we are unable to perform any experiments with non-malignant cells.
-
Thank you very much for your comment. We have extensively discussed it, and have to answer that we had another aim if this study. The role of xCT in ferroptosis development was previously shown in a number of studies (PMID: 33378529; PMID: 32606760; PMID: 30799221; PMID: 33675143, etc. ). Unfortunately, exact molecular mechanisms of erastin-caused ferroptosis should further be elucidated, and we believe that our manuscript contributed to understanding of these intracellular events. Furthermore, the authors suppose that Erk1/2 is phosphorylated as a downstream of the EGFR and independently from the xCT. However, this is a good idea to determine the role of the xCT in our next study.